

# The Nextflow nf-core/metatdenovo pipeline for reproducible annotation of metatranscriptomes, and more

Danilo Di Leo[1,*], Emelie Nilsson[1,*], Arianna Krinos[2], Jarone Pinhassi[1] and Daniel Lundin[1]

[1] Department of Biology and Environmental Science, Linnéuniversitetet, Kalmar, Kalmar, Sweden
[2] Department of Earth, Environmental, and Planetary Sciences, Brown University, Providence, RI, United States of America

[*] These authors contributed equally to this work.

## ABSTRACT

Metatranscriptomics–the sequencing of community RNA–has become a popular tool in microbial ecology, proving useful for both *in situ* surveys and experiments. However, annotating raw sequence data remains challenging for many research groups with limited computational experience. Standardized and reproducible analyses are important to enhance transparency, comparability across studies, and long-term reproducibility. To simplify metatranscriptome processing for biologists, and to promote reproducible analyses, we introduce nf-core/metatdenovo, a Nextflow-based workflow. Nextflow pipelines run on different computing platforms, from standalone systems to high-performance computing clusters and cloud platforms (*e.g.*, AWS, Google Cloud, Azure) and use container technology such as Docker or Singularity to reproducibly provision software. Biologists can access the pipeline using either the command line or the Seqera platform, which provides a web browser-based interface to Nextflow pipelines. Collaborating with nf-core ensures high-quality, documented, reproducible workflows. Our nf-core/metatdenovo pipeline adheres to these established standards, enabling FAIR metatranscriptome *de novo* assembly, quantification, and annotation.

## INTRODUCTION

Over the last two decades sequencing of expressed mRNA from microbial communities—metatranscriptomics (metaT)—has become widely applied in research. From a modest start in 2006, the number of PubMed entries has now reached around 500 yearly mentions of the methodology, and the number is increasing ([Fig. 1]). The method quantifies gene expression and is simultaneously applicable to overall transcription of the communities (*Satinsky et al., 2014*) and groups of taxa or functional genes (*Wang, Gerstein & Snyder, 2009*). The approach has been applied to scientific questions interrogating various biomes, as well as used as the readout for experiments. Examples include a study by *Gilbert et al. (2008)*, into the variation of gene transcripts between mid- and post-bloom communities in a marine environment, the structure of bacterial communities in Arctic peat soil (*Tveit, Urich &*

Corresponding author
Daniel Lundin, daniel.lundin@lnu.se

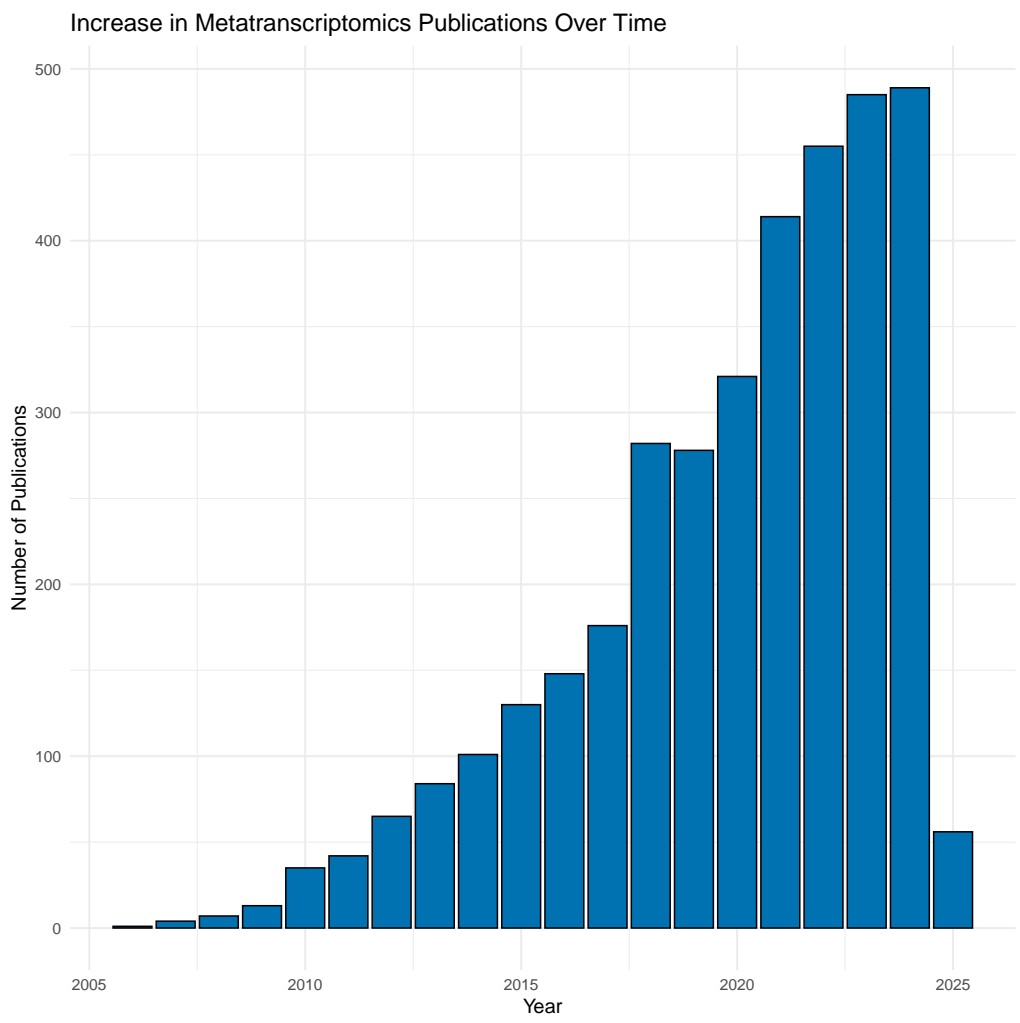

**Figure 1 Results from searching "metatranscriptomics" as keyword on PubMed, Jan 2025.** Each column represents the number of articles published in a given year. As shown in the figure, the number of articles is tremendously increased during time with almost 500 articles in 2024.

*Svenning, 2014*), the role of human gut microbiota in carbohydrate metabolism (*Gosalbes et al., 2011*) and differences in composition and activity of rumen microbiota associated with high or low methane production in cattle (*Neves et al., 2020*). Thus, metatranscriptomic analysis is an approach that interprets how molecular mechanisms function and are regulated, ultimately providing data to guide the assessment into the functioning and health of individual organisms as well as whole ecosystems (*Gallardo-Becerra et al., 2020*; *Pontiller et al., 2022*).

To infer biological meaning from transcripts, it is preferable that both the taxonomy and function of transcripts is established. In the absence of reference genomes to map transcripts to—a situation that is changing thanks to the improvement in recovery of metagenome assembled genomes (MAGs) but still holds for many environments—the analyst has the

benefit of using genome-independent methods. This can be done by aligning trimmed short reads to a database such as RefSeq (*O'Leary et al., 2016*) using computationally efficient alignment programs such as DIAMOND (*Buchfink, Xie & Huson, 2015*) or, for taxonomic classification, k-mer-based approaches such as Kraken2 (*Wood, Lu & Langmead, 2019*). The drawback with read-based annotation is the relatively short length of the reads, usually a maximum of 150 bp for Illumina sequencing, corresponding to a maximum of 50 amino acids. The prediction of community function and taxonomy from short reads is error-prone and computationally expensive (*Pearman, Freed & Silander, 2020*). An alternative approach is to generate a *de novo* assembly of the sequence reads, resulting in longer sequences—contigs—to which taxonomy and function can be assigned with greater precision after gene calling (*Raghavan et al., 2022*). Besides the advantage of increasing the sequence lengths this typically also reduces the number of sequences needed to be aligned to reference databases by orders of magnitude (*Anwar et al., 2019*). The reduced computational cost of this form of investigation is somewhat negated by the cost of assembly and quantification of the resulting contigs and features, typically performed by mapping reads back to an assembly (*Raghavan et al., 2022*).

The large number of steps involved in the *de novo* assembly of metaT data makes the adoption of the method difficult for research groups with limited access to computational expertise. In addition, reproducibility poses a challenge as many computational tools are involved, and the lack of standards in data processing is a recognized problem in the scientific community (*Pinhassi et al., 2022*). Technologies that can assist in making data processing more accessible with improved reproducibility are thus crucial. Standardized pipelines, written for workflow managers (*Wratten, Wilm & Göke, 2021*), such as Nextflow (*Di Tommaso, Chatzou & Floden, 2017*) or Snakemake (*Mölder et al., 2021*), promise to fill this need (*James, 0000*). In contrast to pipelines written in Bash or other scripting languages that typically depend on locally installed software and computational resources, workflow managers provide software, of specific versions, using package managers such as Conda (*Grüning, Dale & Sjödin, 2018*) or, preferably, container software such as Docker (*Merkel, 2014*) or Singularity (*Kurtzer, Sochat & Bauer, 2017*). Moreover, through configuration options, they allow the execution of pipelines on a wide array of hardware platforms, from laptop and desktop computers *via* high-performance computing clusters to cloud-based environments, allowing research groups to choose the best option for their project and budget (*Wratten, Wilm & Göke, 2021*). In the case of analysis failures or those analyses aborted for other reasons, pipelines can be restarted without having to recompute results. In conclusion, workflow managers promise to enhance the accessibility and reproducibility of computational data analysis.

It is important to mention that software, including pipelines, are maintained so that errors are corrected and improved tools are continuously made available (*Spjuth et al., 2015*). Unfortunately, the scientific literature contains reference to numerous tools that have not been maintained since their publication (*Di Tommaso, Chatzou & Floden, 2017*; *Leipzig, 2016*). Although this is an inherent problem related to non-commercial open source development, projects that are shared with a larger community have arguably a better chance of surviving, particularly if combined with implementation standards and

peer review (*James, 2022*). The nf-core collaboration (*Ewels et al., 2019*) is an example of such a community. At the time of writing, it maintains 83 released Nextflow pipelines—the vast majority supporting bioinformatics—with 42 under development. Nearly 2,500 researchers have contributed to the pipelines, a fifth of whom have made code commits in at least one of the pipelines' GitHub repositories. The goal of the nf-core collaboration is to enhance the longevity and reliability of bioinformatics software.

This article introduces nf-core/metatdenovo, a pipeline designed for fast and straightforward *de novo* assembly, quantification, taxonomic classification, and functional annotation of metatranscriptomes. Although nf-core/metatdenovo was initially developed for metatranscriptomic datasets, it is also applicable to metagenomic (metaG) data, offering different assembly options appropriate to each data type. To be noted, however, is that the pipeline does not include binning steps to reconstruct metagenome-assembled genomes (MAGs). Instead, the pipeline focuses on assembly and annotation of contigs, providing taxonomic and functional information resulting in annotated and quantified gene catalogs. (For the reconstruction of MAGs, see the nf-core/mag pipeline (*Krakau et al., 2022*)). In principle, it is also possible to combine metaT and metaG data. The presented pipeline uses Nextflow (*Di Tommaso, Chatzou & Floden, 2017*), forms part of the nf-core collaboration (*Ewels et al., 2019*) and is released under the permissive Massachusetts Institute of Technology (MIT) license. We describe the implementation of the pipeline and evaluate its performance on three previously published datasets to showcase its utility for different types of datasets. One dataset was originally published with a manually performed assembly-based annotation and consists of an experiment performed on natural marine prokaryotic communities (*Bunse et al., 2016*). A second prokaryotic dataset was taken from a fermentation process and consists of communities of low diversity and was originally annotated by mapping to the genomes of a few dominant community members (*Jung et al., 2013*). To investigate the pipeline's versatility on other types of communities, we also applied nf-core/metatdenovo to a project consisting of predominantly eukaryotic communities (*Alexander et al., 2015*). Finally, we provide a qualitative comparison with other published pipelines that overlap with nf-core/metatdenovo: Eukrhythmic (*Krinos et al., 2023*), MetaGT (*Shafranskaya et al., 2022*), SAMSA2 (*Taj et al., 2023*) and MetaPro (*Westreich et al., 2018*). These pipelines diverge in processing strategy: Eukrhythmic, MetaGT, MetaPro, and nf-core/metatdenovo use a *de novo* assembly to create longer contigs for downstream analysis whereas SAMSA2 employs an assembly-free approach, by mapping raw reads to databases. Furthermore, Eukrhythmic is implemented in Snakemake, MetaGT and nf-core/metatdenovo uses Nextflow, while MetaPro and SAMSA2 operate as standalone, but containerized, scripts.

## MATERIALS & METHODS

### nf-core/metatdenovo: pipeline description
#### Pre-processing steps
The nf-core/metatdenovo pipeline annotates sequence data as follows (Fig. 2): The raw reads are quality controlled and trimmed with Trim Galore! *Krueger et al. (2023)* and, if
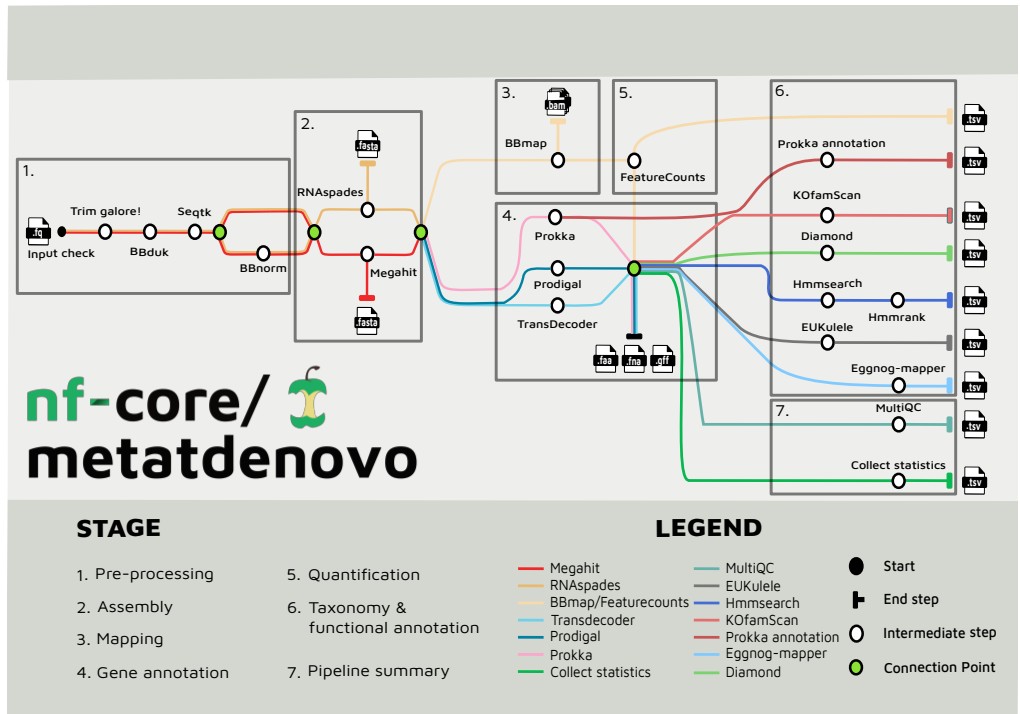

**Figure 2** **Tube map of nf-core/metatdenovo.** The tube map is showing all the available tools in nf-core/metatdenovo and how they are linked to each other. Different colours represent different workflows which end in a formatted table.

the user supplies a file with unwanted sequences—*e.g.,* stable RNA and/or host genome sequences—the reads are then filtered with BBDuk (*Brian, 2014*). Subsequently, Seqtk (*Heng, 2016*) is used to interleave paired reads. After this step, sequence read depth can be normalized using BBNorm (*Brian, 2014*). Read depth in metatranscriptomics samples often varies considerably due to the dual effects of variable organism abundance and unequal transcriptional activity between genes within organisms. By reducing excess read depth from overrepresented sequences, normalization facilitates assembly, potentially improves the quality of the assembly, and reduces computational resource usage.

### Assembly and gene annotation steps

Assembly of quality-controlled and potentially filtered and normalized sequence reads can subsequently be performed either with SPAdes (*Bushmanova et al., 2019*; *Prjibelski et al., 2020*) or MEGAHIT (*Li et al., 2015*). The former is a multi-purpose assembler with modes for optimization for different tasks. In nf-core/metatdenovo, the default mode is "rnaspades", but other modes are available using the—spades_flavor parameter (see the online documentation (https://nf-co.re/metatdenovo)). MEGAHIT was originally designed to assemble metagenomes, but has successfully been applied also to metaT data (*Raghavan et al., 2022*). Nf-core/metatdenovo provides three gene calling options to accommodate preferences: two for prokaryotes—Prodigal (*Hyatt et al., 2010*) and Prokka (*Seemann, 2014*)—and one for eukaryotes—TransDecoder (*Haas, 2015*). The difference between

the two prokaryotic options is not the Open Reading Frame (ORF) caller *per se*, which is Prodigal in both cases, but Prokka, which, in addition to calling ORFs, identifies other types of transcripts, *e.g.*, rRNA and tRNA, and performs similarity-based functional annotation. TransDecoder detects and scores ORFs within transcript sequences and does not attempt to identify non-coding RNAs or perform functional annotation as Prokka. Since microbial communities often are composed of both prokaryotes and eukaryotes, the workflow can run a combination of the three gene calling options. All subsequent steps will be run for each of the gene callers, and separate output files will be outputted.

### Quantification steps

In parallel to gene calling, the first step of quantification is performed by mapping the quality-controlled and filtered, but not depth-normalized, reads back to the assembly using BBMap (*Brian, 2014*). Quantification of gene expression is then finalized with featureCounts from the Subread package (*Liao, Smyth & Shi, 2014*) to retrieve the counts for each ORF from the assembly in each sample.

### Functional annotation steps

In addition to the possibility of using Prokka as an ORF caller and functional annotator, other functional annotation programs are available in the workflow. eggNOG-mapper (*Cantalapiedra et al., 2021*) or KofamScan (*Takuya, 2020*) can be used to annotate ORFs using the eggNOG (*Huerta-Cepas et al., 2019*) or KEGG (*Minoru, 2002*) annotation systems, respectively, each providing a general-purpose similarity-based annotation. Moreover, users interested in specific genes that are not well recognized by the general annotation tools available in the workflow can provide amino acid HMMER (*Finn et al., 2015*) profiles that are used to search the translated sequences of the ORFs.

### Taxonomy annotation steps

The ORFs from the assembly can be taxonomically annotated using EUKulele (*Krinos et al., 2021*) or DIAMOND (*Buchfink, Xie & Huson, 2015*). EUKulele bases its annotations on a selection of databases, all with their own set of organism genomes annotated and taxonomy recorded: MMETSP (marine eukaryotic transcriptomes) (*Keeling et al., 2014*), MarRef combined with MMETSP (marine bacteria combined with marine eukaryotic transcriptomes), PhyloDB (a selection of both prokaryotes and eukaryotes) (*A. E. Allen Lab, 2015*), EukProt (Eukaryotes) (*Richter et al., 2022*), and GTDB (*Parks et al., 2022*) for prokaryotes. See the EUKulele documentation: https://eukulele.readthedocs.io/en/latest/databaseandconfig.html. In DIAMOND, the user can provide any database that has been formatted to contain taxonomic information. We provide instructions for formatting a database in the online documentation of nf-core/metatdenovo as well as a preformatted GTDB (release RS09-R220) at FigShare: https://figshare.scilifelab.se/articles/dataset/nf-core_metatdenovo_taxonomy/28211678.

### Pipeline summary

The output files of the tools described are saved in their native formats, in directories named by default by the software. In addition, the workflow produces tab-separated tables

summarizing the output from all tools, with consistent naming of fields, ready for analysis in programming languages like R and Python. Finally, to describe the overall performance of the pipeline, MultiQC (*Ewels et al., 2016*) is used to produce an HTML-formatted report of pipeline performance, and an R script is called to produce a tab-separated file summarizing overall statistics from trimming, assembly, mapping, annotation, *etc.*

## Dataset descriptions

To demonstrate the functionality of the pipeline, we re-analysed three different datasets from published article. We chose two datasets for which we have extensive experience as authors of the original publications: MST-1, a mesocosm dataset with bacterioplankton surface water communities (*Alexander et al., 2015*) and a study of mesocosm phytoplankton acidified activity (*Bunse et al., 2016*), and a third dataset on the microbiome of kimchi fermentation (*Jung et al., 2013*).

## Marine bacterioplankton dataset

The MST-1 dataset was generated from a mesocosm experiment analysing the surface water from the Western Mediterranean, where *Bunse et al. (2016)* established the impact of acidification on homeostasis genes in the bacterial community. In the experiment, the influence of nutrient addition and acidification was tested in a full factorial design with treatments performed in duplicate. Nutrient-amendment triggered a phytoplankton bloom, which led to an increase in chlorophyll a concentration in the samples. These samples were denoted "High-Chl-acidified" and "High-Chl-control" respectively for the acidified and non-acidified treatments, whereas the non-nutrient-amended samples were denoted as "Low-Chl-acidified" and "Low-Chl-control" respectively. *Bunse et al. (2016)* found a strong response to acidification in the "Low-Chl-acidified" treatments, and a much weaker response in the "High-Chl-Acidified" treatments. As Bunse et al. explain in the paper, metatranscriptomic sequencing and annotation was performed with *de novo* assembly (Velvet (*Zerbino & Birney, 2008*) and Ray (*Boisvert, Laviolette & Corbeil, 2010*)) after quality trimming and removal of stable RNA transcripts (Erne (*Prezza et al., 2016*)). ORFs were called with FragGeneScan. The ORFs were then functionally and taxonomically annotated using BLAST (*Camacho et al., 2009*) against the M5NR SEED, KEGG, and RefSeq databases using a script developed in-house. Their pipeline annotated between 19 and 51% of reads. The quantification was carried out using Bowtie2 (*Langmead & Salzberg, 2012*).

We downloaded the raw sequence reads from ENA (PRJEB10237) to re-analyze the MST-1 dataset with nf-core/metatdenovo and ran pipeline (v. 1.1.1) using eight combinations of the parameters the pipeline provides in order to showcase the consequences of different parameter choices. The initial steps of the pipeline—quality checks, trimming, filtering out rRNAs—were identical in all downstream processing steps. After these steps, we ran read depth normalization to provide input reads for assembly. Subsequently, the assembly was run with either of the two available assemblers (MEGAHIT and SPAdes, using the default rnaSPAdes mode) using the full set of reads or the read-depth normalized set, resulting in four assemblies. Each of the assemblies was ORF called with both of the options suitable for prokaryotic genes using Prodigal and Prokka. Finally, to test the influence of

the "–diamond_top" parameter, we reran the MEGAHIT/Prokka run with this set to 3 (10 being the default), resulting in a total of nine runs and sets of ORFs. To allow better comparisons between the two assemblers, we ran ORF calling on the rnaSPAdes assembly using the pipeline option "–min_contig_length" set to 200, since MEGAHIT by default filters contigs by this length. For this project, the pipeline was run on an AMD Ryzen 9 3900X 12-core processor, using hyperthreading for 24 cores in total, and 128 GB RAM, with Docker as the container technology.

## 'Kimchi' microbiome

The 'Kimchi' dataset consists of five metatranscriptome samples obtained during a 29-day kimchi fermentation process, collected on days 7, 13, 18, 25, and 29 (SRA050204). *Jung et al. (2013)* published a study in which they analyzed the homofermentative process of a selected fermenting community. Based on prior knowledge, they annotated their metatranscriptomic reads by mapping to genomes of known representatives of the community. The article documented the dynamic shifts in these populations during fermentation, investigating distinct functions associated with each day, particularly focusing on enzyme production.

For our comparison, we downloaded the raw sequence reads SRX128705 (J29), SRX128704 (J25), SRX128702 (J19), SRX128700 (J13), SRX128699 (J7), and ran nf-core/metatdenovo (v. 1.0.1) with MEGAHIT, using Prokka for ORF calling. We used the Silva rRNAref_NR99 database to remove rRNA (release 138; (*Quast et al., 2013*)). ORFs were called using Prokka. The annotation was using EggNOG-mapper for function and EUKulele (GTDB; release RS07-R207) for taxonomy. Our objective was to demonstrate the feasibility of addressing the same research question with *de novo* assembly-annotated reads.

## Eukaryotic dataset

The eukaryotic dataset was published by *Alexander et al. (2015)* and reanalyzed by *Krinos et al. (2023)*. nf-core/metatdenovo (v. 1.0; modified to use EUKulele 2.1.0) was used to assemble and annotate samples S1, S2, S3, S4, and S5 from the article, which corresponds to samples from May 16, May 21, May 30, June 4, and June 8, 2012 (*Alexander et al., 2015*). In the original article, these samples were analyzed by direct mapping of reads to the MMETSP, with a specific focus on two phytoplankton taxa of particular interest (the diatoms *Skeletonema costatum* and *Thalassiosira rotula*). In *Krinos et al. (2023)*, a combined assembly approach was applied to the same dataset using the Eukrhythmic pipeline, the EUKulele tool for taxonomic annotation (v. 2.1.0), and the eggNOG-mapper tool for functional annotation. We downloaded the raw reads from the SRA to run within nf-core/metatdenovo and to compare the weighted abundance of taxa of interest using a similar methodology to *Krinos et al. (2023)* *via* the results obtained using EUKulele with the same reference database (the MarRef database (*Klemetsen, 2021*) and the MMETSP database, combined as MarMMETSP v. 1.0). We then compared the results obtained by nf-core/metatdenovo to the results of the original paper using direct read mapping and the Eukrhythmic pipeline (*Krinos et al., 2023*). SRA datasets SRR1810799
(S1), SRR1810204 (S2), SRR1810801 (S3), SRR1810205 (S4), and SRR1810206 (S5) were used. As in the comparison to *Bunse et al. (2016)* above, we did not expect that the results of nf-core/metatdenovo would be fully comparable to the raw read mapping approach adopted by *Alexander et al. (2015)* due to methodological differences. However, it was expected that the assembly-based approach should be directly comparable to *Krinos et al. (2023)*, though a single assembler (MEGAHIT v. 1.2.9) was used in the nf-core/metatdenovo pipeline while MEGAHIT, rnaSPAdes, metaSPAdes, and Trinity were used in combination in the Eukrhythmic pipeline (*Krinos et al., 2023*). We evaluated the outputs of nf-core/metatdenovo by comparing the breakdown of key taxonomic annotations from EUKulele to both the raw read mapping approach and to Eukrhythmic and compared the proportion of raw reads mapping to the assembly between nf-core/metatdenovo and Eukrhythmic.

### Data visualization and code availability

Data visualization was performed with R v. 4.3.2 (*R Core Team, 2023*) and packages in Tidyverse (*Wickham et al., 2019*). The Quarto documents for MST-1 and Kimchi are available in the repository https://github.com/LNUc-EEMiS/metatdenovo-paper, under 'mst-1_analyses.qmd' and 'kimchi-analysis.qmd'. All files needed to run the scripts are available in the repository. A table listing all software used in nf-core/metatdenovo is available in Tables S1 and S2.

## RESULTS

### Pipeline performance on the marine bacterioplankton dataset
*Resource usage and assembly quality with different parameter values*

With the *Bunse et al. (2016)* dataset consisting of surface water prokaryotic communities from a mesocosm experiment, we tested how different parameter choices for the assembly software and ORF callers, as well as if using read depth normalization, influence the quality of the assembly and the recruitment of ORFs. The MST-1 dataset is, by today's standards, relatively small, and resource usage was moderate for all the configurations tested (Fig. 3). Around a third of the computational time expended by the workflow was used in the early steps of the workflow—quality trimming, removal of rRNA reads with BBDuk—and another third with quantification by mapping reads back to the assembly (Fig. 3A). The assembly step used a relatively small proportion of computing time (up to 10%). Memory usage differed considerably, however, as MEGAHIT used less than half of the resources that SPAdes did (Fig. 3B). In fact, it used less memory on the full read set than SPAdes used for the depth-normalized set. ORF calling with Prodigal was close to instantaneous compared to the other processes, while with Prokka it was amongst the most time-consuming steps, using between 10% (MEGAHIT-Prokka+BBnorm) and 29% (SPAdes-Prokka without BBnorm) of the total runtime. This is consistent with Prokka being more than an ORF caller, as it also searches for other features in addition to protein-coding genes and adds annotations to features. Finally, taxonomic and functional annotation used relatively small proportions of the total time (approximately 10%).
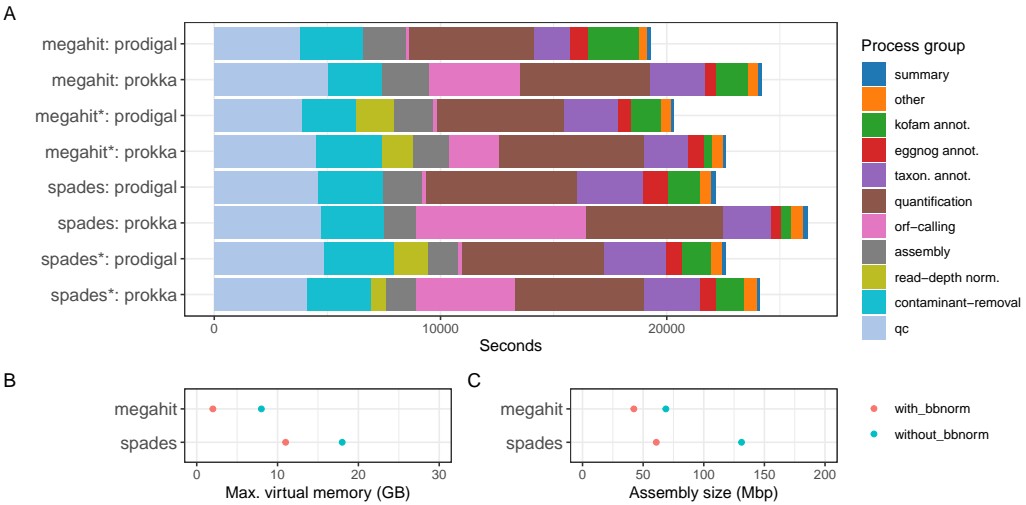

**Figure 3  CPU consumption, assembly memory usage, and size.** (A) Real-time CPU use per task type and pipeline execution, (B) peak virtual memory usage, and (C) assembly size per assembly. (*) The assembly was made from read-normalized data. For Spades-assemblies, contigs shorter than 200 nucleotides were removed using the –min_contig_length parameter.

What the results in Fig. 3 do not directly take into account is, firstly, that the complexity and the size of the dataset influence the relative proportions of time taken by each step. Secondly, certain steps, *e.g.*, quality trimming and mapping, are inherently parallelizable since they are applied to every sample, while others, *e.g.*, assembly and annotation, are not. Running the pipeline on parallel hardware, like high-performance computing (HPC) clusters or cloud providers, can strongly affect the wall time of the analysis. Choice of assembler, and whether to read-depth normalize the input reads presents some trade-offs. For a dataset like MST-1 that can easily be analyzed on moderately powerful hardware, our analysis indicates that the choice of assembler should mainly be informed by other characteristics of the assemblies—such as total size, N50 or other length measures, or the proportion of reads mapping back to the assembly—than the resource consumption (Fig. 3A). For larger projects, the smaller resource usage of MEGAHIT—particularly memory—would dictate the choice. Moreover, although read-depth normalization resulted in considerably lower proportions of reads mapping back to the assembly (Table 1), this may be a useful strategy for very large projects. SPAdes generated larger but more fragmented assemblies with a higher proportion of mapping reads (Table 1).

### ORF calling and annotation

Prokka returned far fewer ORFs than Prodigal (62,769 Prokka ORFs and 130,146 Prodigal ORFs in the MEGAHIT assembly, and 78,737 and 298,304, respectively, in the SPAdes assembly, both from non-normalized reads). This was expected, given the more conservative functioning of Prokka. The SPAdes-Prodigal combination identified a large number of rare ORFs found in a few samples (Fig. 4A). It also produced ORFs with a slightly shorter length distribution (Fig. 4C). While it is difficult to determine which ORFs are real,

**Table 1** Overall statistics for MST-1 assemblies.

| Assembly | N. contigs | Size (Mbp) | Lengths | Mean length (bp) | N50 | % mapped |
|----------|-----------|-----------|---------|-----------------|-----|----------|
| megahit* | 48,826 | 42.3 | 200-67233 | 867 | 1,108 | 62.3% |
| megahit | 97,522 | 68.7 | 200-103660 | 705 | 776 | 68.6% |
| rnaspades* † | 85,886 | 60.8 | 200-50341 | 713 | 708 | 72.1% |
| rnaspades† | 262,652 | 131.2 | 200-59389 | 500 | 499 | 78.8% |

**Notes.**

*The assembly was made from read-normalized data.

†For Spades assemblies, contigs shorter than 200 nucleotides were removed using the -min_contig_length parameter. MEGAHIT does not output contigs shorter than 200 nucleotides.

it is essential to acknowledge that Prodigal's less stringent approach increases the risk of detecting false positive ORFs. For instance, in the MEGAHIT assembly, Prodigal identified approximately 7,283 ORFs with a length shorter than 90 amino acids, while Prokka did not identify any such ORFs. Read-depth normalization in general produced assemblies which after ORF calling have a much narrower count distribution than the corresponding non-normalized assembly (Fig. 4A). The proportions of reads mapping back to the ORFs and the relative proportions between samples was strikingly similar between the assembler/ORF caller combinations, with or without read-depth normalization suggesting that abundant transcripts are well covered by any parameter choice (Fig. 4B).

### Annotation success

There are two annotation tools for taxonomy supported by the pipeline: EUKulele (*Krinos et al., 2021*), present since v. 1.0 of the pipeline, and DIAMOND (*Buchfink, Xie & Huson, 2015*) since v. 1.1. Both tools support different databases, and were run with the GTDB database, but with different versions: R09-RS220 for DIAMOND and R07-RS207 for EUKulele. In addition, the DIAMOND tool was run with an NCBI RefSeq database, which was used to filter out eukaryotic sequences. Differences between the assembler/ORF caller combinations were clear in terms of the number of taxonomically annotated ORFs, but small when weighting the ORFs either by the raw read pair counts or transcripts per million (ORF length-weighted relative abundances; TPMs) (Fig. 5A). The precision of assignment differed considerably between the two tools, with EUKulele achieving much higher proportions of assignments at species level than DIAMOND. For the DIAMOND tool, the range of alignment scores to consider for taxonomy assignment can be tuned with the –diamond_top parameter, set to 10 by default, *i.e.,* a 10% range from the best scoring alignment. Setting this to 3, assigned several more ORFs at the species level than the default (Fig. 5A). The functional annotation of ORFs in nf-core/metatdenovo uses the Prokka output, as well as the eggNOG-mapper (*Cantalapiedra et al., 2021*) or KofamScan (*Takuya, 2020*). Cross-comparison of these annotation programs revealed that the majority of ORFs were annotated by all tools run, *i.e.,* either all three or, in the case when Prodigal was used to call ORFs, both the eggNOG-mapper and KofamScan (Fig. 5B). The eggNOG-mapper stands out as a particularly robust standalone program, annotating almost all ORFs, alone
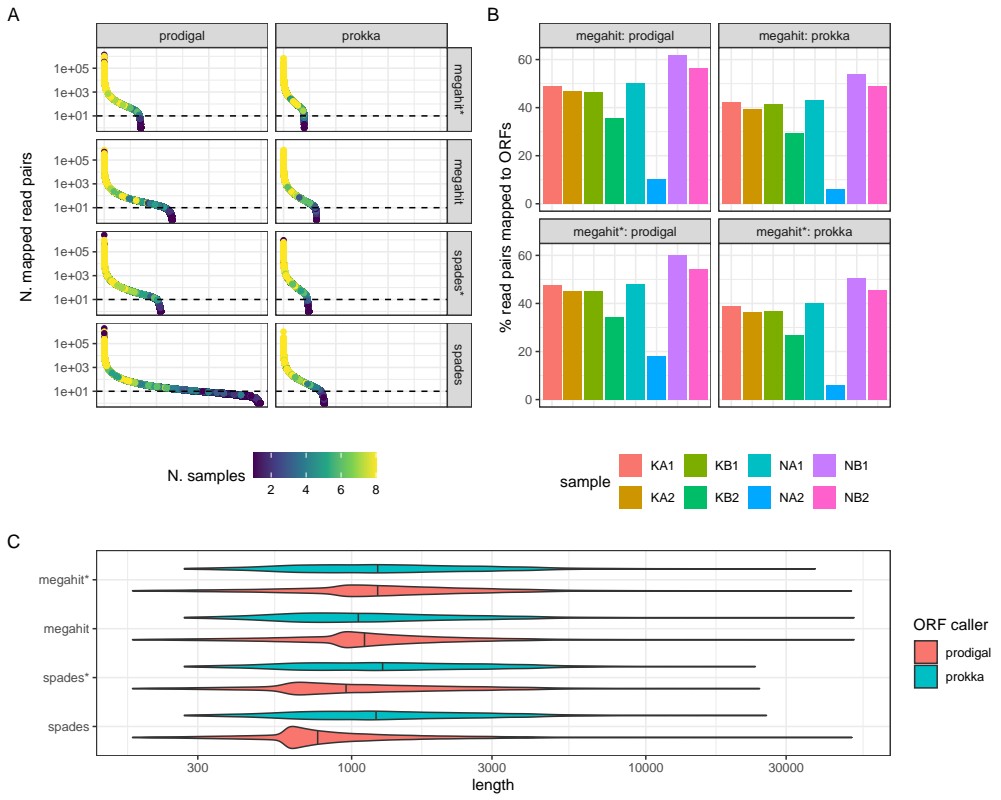

**Figure 4 ORF calling and mapping statistics.** (A) Rank abundance distributions of ORFs in different combinations of assembly programs and ORF callers. Counts were summed over all single sums per ORF. The color scale shows the number of samples in which each ORF was detected. Note the $y$-axis is a log scale. The $x$-axis represents the ordering of ORFs in falling abundance. (B) Percentages of read pairs mapping to ORFs in each sample. (C) Length distributions of ORFs for the different combinations of assemblies and ORF callers. Note the log scale. (*) The assembly was made from read-normalized data. For Spades-assemblies, contigs shorter than 200 nucleotides were removed using the -min_contig_length parameter.

or in combination with the other tools, whereas KofamScan and Prokka annotated slightly smaller proportions of ORFs.

### Comparison of our results with the published study

Both the original study by *Bunse et al. (2016)* and nf-core/metatdenovo annotate the raw reads by first assembling them. However, the assembly strategy differed as nf-core/metatdenovo generates a single assembly from all samples, whereas Bunse et al. performed two per-sample assemblies using different assembly programs. To arrive at a single set of ORFs after assembly and ORF calling, ORFs were clustered, quantified, and annotated against a set of databases, both for taxonomic and functional assignment. Bearing in mind the methodological differences between the assembly-based nf-core/metatdenovo pipeline and the annotation strategy used in the *Bunse et al. (2016)* study, we here compare the main findings of the two approaches. For this comparison, we chose the

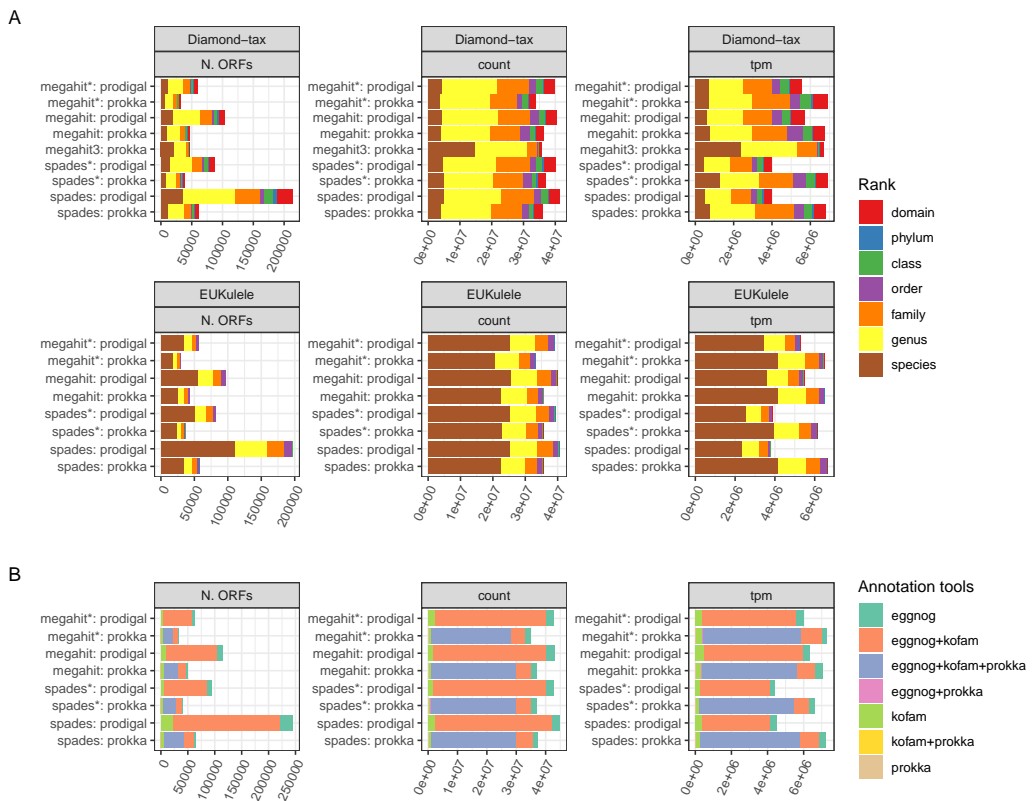

**Figure 5** **Annotation success.** (A) GTDB taxonomy with the DIAMOND and EUKulele tools; and (B) combinations of functional annotation. Unweighted, weighted by count, and tpm, respectively. The taxonomy annotations exclude ORFs annotated as eukaryotic by the DIAMOND tool using NCBI RefSeq as the database. Proteins annotated as "hypothetical protein" by Prokka were not included. (*) The assembly was made from read-normalized data. †) Contigs shorter than 200 nucleotides were removed using the -min_contig_length parameter. (3) DIAMOND taxonomy run with the –diamond_top set to 3 instead of its default 10.

nf-core/metatdenovo run using non-normalized reads, MEGAHIT as the assembler, and Prokka as the ORF caller. As this dataset mostly consisted of prokaryotes, we chose the GTDB database to define taxonomy (using the EUKulele annotation program), which we here compare to the NCBI taxonomy used by *Bunse et al. (2016)* (Fig. 6A). The most striking difference between nf-core/metatdenovo and the original study's taxonomy annotation was that the *Flavobacteriaceae*, *Alteromonadaceae*, and *Rhodobacteraceae* families reached much higher transcript abundances in the nf-core/metatdenovo annotation compared to the original study. In the cases of *Alteromonadaceae* and *Rhodobacteraceae*, this was partly compensated by larger proportions of the corresponding "Other Alphaproteobacteria" and "Other Gammaproteobacteria" categories in Bunse et al., likely due to higher precision in the nf-core/metatdenovo annotation. This was, however, not the case with the corresponding flavobacterial categories, possibly pointing to poor identification of these taxa even at the class level in the NCBI annotation in the original approach. Overall, the total relative abundance of these taxa was also higher in the nf-core/metatdenovo output.

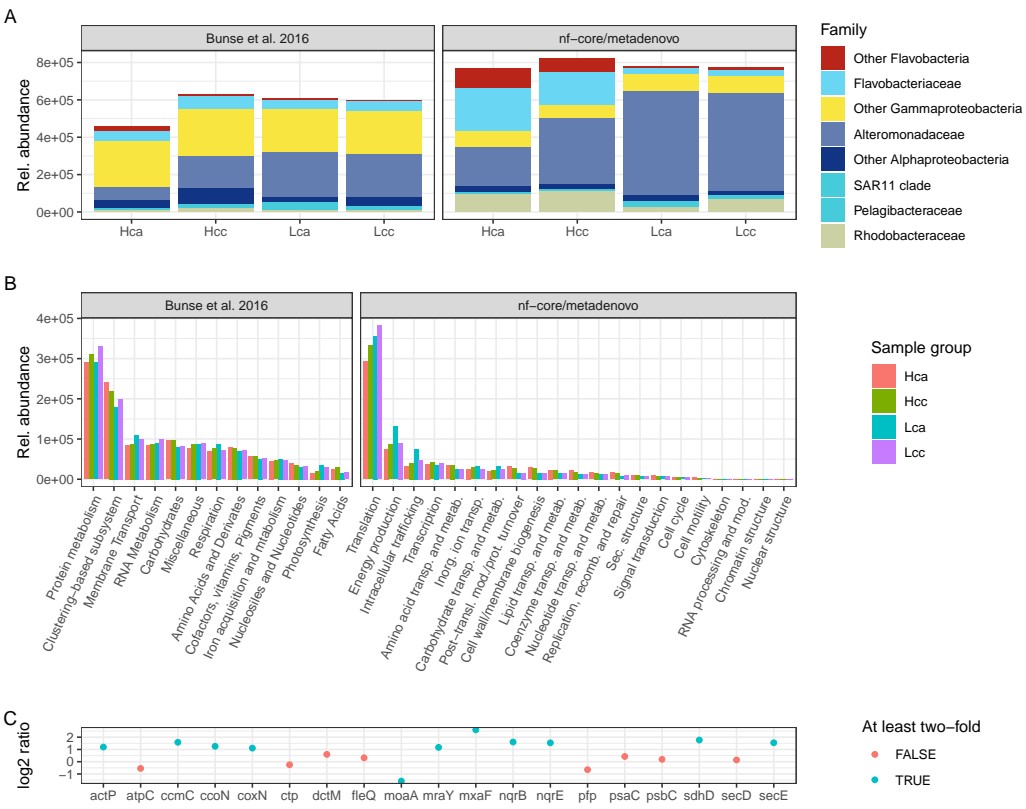

**Figure 6** **Taxonomic and functional annotation comparison.** (A) The most abundant taxonomic groups in *Bunse et al. (2016)* and nf-core/metatdenovo. NCBI taxonomy for *Bunse et al., 2016* and EUKulele/GTDB RS07-R207 for nf-core/metatdenovo. (B) Functional categories. SEED for *Bunse et al., 2016* and COG categories for nf-core/metatdenovo. (C) Log2 ratio between mean expression in acidified compared to non-acidified low-chlorophyll samples in the nf-core/metatdenovo annotation of genes significantly more abundant in the original study. Only these eleven gene names were possible to identify in the eggNOG-mapper annotation. "Rel. abundance" refers to CPMs (counts per million) in the original study and TPMs (gene length-corrected relative abundances) in nf-core/metatdenovo output. Treatment abbreviations: Hca, High chlorophyll acidified; Hcc, High chlorophyll non-acidified; Lca, Low chlorophyll acidified; and Lcc, Low chlorophyll non-acidified. In high chlorophyll samples were amended with nutrients, causing a phytoplankton bloom. Numbers from *Bunse et al. (2016)* were measured in Fig. 1E and Fig. S1 for taxonomy and function, respectively.

This is likely an effect of both the longer sequences annotated by nf-core/metatdenovo as a result of assembly, and the high specificity achieved by EUKulele in combination with GTDB discussed above.

The original study used the hierarchical SEED system for functional annotation, whereas nf-core/metatdenovo supports functional annotation with the eggNOG-mapper, KofamScan, and Prokka. For a high-level comparison of gene expression, the COG categories provided by the eggNOG-mapper in nf-core/metatdenovo are the closest equivalent to the top level of the SEED hierarchy. In the original study, the authors found protein metabolism to be the most highly expressed group, at $3 \times 10^5$. This was mirrored in the nf-core/metatdenovo output, by the most highly expressed COG category being

Translation at $\sim3$–$4 \times 10^5$ TPM (Fig. 6B). The second most abundant category in the nf-core/metatdenovo annotation was Energy metabolism at $\sim1 \times 10^5$ TPM, potentially corresponding to the 'Respiration and Photosynthesis' categories in SEED summing to $\sim1 \times 10^5$ CPM in the Bunse et al. study.

A key functional finding in the original study was the significantly higher abundance of certain genes in the acidified low-chlorophyll samples. It proved difficult to replicate these results with the nf-core/metatdenovo annotation, highlighting the importance of reproducible annotation workflows. A specific challenge was to map the gene names published in *Bunse et al. (2016)* to the nf-core/metatdenovo annotation. We managed to identify only 19 of the original 47 significant genes reported in the study. Out of these 19, ten were at least two-fold more abundant and one was more than half as abundant in acidified samples compared to non-acidified controls (Fig. 6C).

## 'Kimchi' fermentation dataset

To determine if our *de novo* assembly-based approach works well on data from microbial communities with reduced diversity and genome complexity, and recovers patterns similar as the original study, we ran nf-core/metatdenovo using MEGAHIT as an assembler and Prokka as ORF caller on the reads from *Jung et al. (2013)* (see 'Materials and Methods' for details). We mapped considerably higher proportions of reads to the contigs of the assembly (mean 77.5% of reads after quality check and removal of rRNA genes), compared to the original study's mapping rate (mean 44.1%) (Table S3). By analysing only those reads that mapped to assigned taxonomy, a large part of the differences in mapping rate could be attributed to sample J7 that had a much higher mapping percentage to our assembly than to their genomes ($\sim23\%$ compared to 2%), because the community in this sample was more diverse than the other samples and did not match the reference genomes (Fig. 7A). It is assumed that, at this time, the kimchi fermentation culture had not yet fully developed. The highest mapping score for *Jung et al. (2013)* was observed in J18 with 78%, compared to our result of $\sim60\%$. These results stem from the approach taken by nf-core/metatdenovo, which assumes less about community composition as it starts by building a *de novo* assembly.

Despite the differences in mapping rate, when we focus on targeted genomes, the two approaches were strikingly similar (Fig. 7B). *Leuconostoc mesenteroides* demonstrated a relative expression covering over 50% of the total during J7, then a decline in J18 and a stabilization around 10% for the remaining time. *Weissella koreensis* displayed a gradual ascendancy, becoming prominent during J25. *Lactobacillus sakei* emerged as the dominant species among the six by J29—a consensus observed in both analyses. *L. inhae* (previously *Leuconostoc gasicomidatum*) shifted from a higher concentration in J7 to an almost absence by J29 (Fig. 7B).

To test functional annotation's performance, we identified the target genes from the original study—involved in the homofermentative pathways—in the nf-core/metatdenovo output using the information provided by Prokka and eggNOG-mapper to see if we could find a similar pattern as the original article. Because of the lower taxonomic precision of nf-core/metatdenovo, we chose to study the expression of the three genera represented

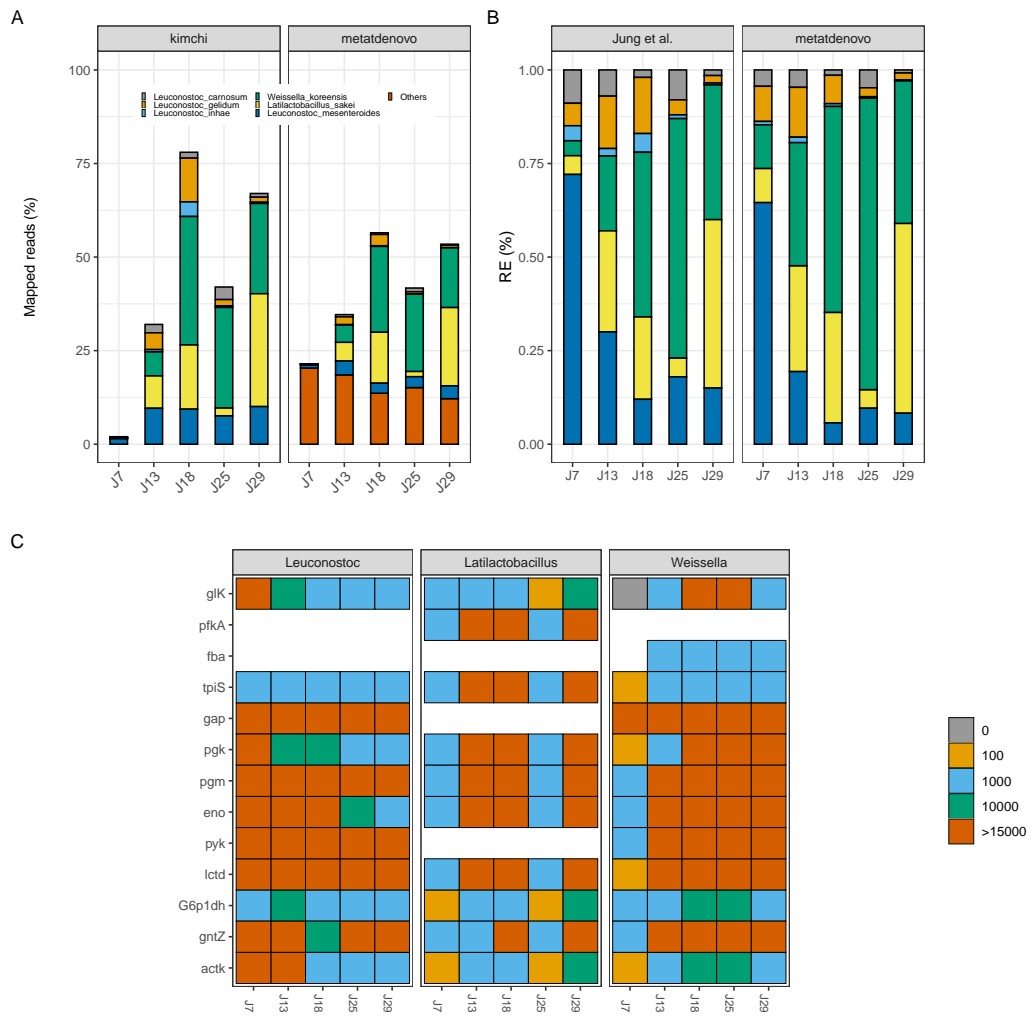

**Figure 7** **Kimchi dataset comparison.** (A) Percentage of mapped reads, a comparison between *Jung et al. (2013)* annotation (left) and nf-core/metatdenovo output (right). The colors are the same, the last one includes "other" taxa. (B) Taxonomic comparison of relative gene expression between kimchi original paper (left) and nf-core/metatdenovo (right). (C) Three selected genera are chosen to show how key enzymes are expressed during a homofermentative process. The expression level changes from blue, low expression, to red, high expression. It is possible to appreciate the shift in enzyme expression and in the involvement of the different genera over the progression of the experiment. Since assigning genes to species is not always possible in a *de novo* assembly of a transcriptome, we chose to study this at the genus level instead of species. From the top glucokinase, ATP-dependent 6-phosphofructokinase, aldolase, triosephosphate isomerase, glyceraldehyde-3-phosphate dehydrogenase, phosphoglycerate kinase, phosphoglycerate mutase, enolase, pyruvate kinase, lactate dehydrogenase, glucose-6-phosphate 1-dehydrogenase, 6-phosphogluconate dehydrogenase, acetate kinase.

instead of species. We observed similar trends compared to the *Jung et al. (2013)* article both in terms of who is expressing what and when, as *Latilactobacillus* in J13, J18 and J29 but also in what they are expressing as key genes glucokinase and 6-phosphogluconate dehydrogenase expressed by *Weissella* genus (Fig. 7C). The two very different annotation approaches arrived at similar conclusions for the community evolution as well as the

**Table 2 Percentage mapping statistics returned by Eukrhythmic (columns 2), nf-core/metatdenovo (column 3), and the read mapping approach from *Alexander et al.* (*2015*; column 4).** While Eukrhythmic uses the Salmon tool against the nucleotide assembly, nf-core/metatdenovo uses the Subread program on predicted ORFs. For all assemblies except S1, the rank order of percentage mapping numbers was similar between Eukrhythmic and nf-core/metatdenovo.

| Sample | Percent mapped-eukrhythmic/ Salmon | Percent mapped-metatdenovo/ Subread | Percent mapped-*Alexander et al., 2015* approach |
|---|---|---|---|
| S1 | 79.3% | 63.3% | 78.3% |
| S2 | 77.5% | 70.7% | 75.1% |
| S3 | 80.3% | 73.5% | 67.8% |
| S4 | 84.1% | 73.3% | 63.4% |
| S5 | 86.9% | 79.4% | 62.5% |

expression patterns during fermentation. Altogether, our results show that it is possible through nf-core/metatdenovo to discover key taxa related to kimchi fermentation and their key enzymes involved in the process.

### Eukaryotic dataset from Narragansett Bay

For the eukaryotic study analysing genomes sequences from Narragansett Bay, we compared the results of running nf-core/metatdenovo both to the original raw read mapping-based approach (*Alexander et al., 2015*) and the results of the Eukrhythmic pipeline (*Krinos et al., 2023*). We compared the number of reads mapped to contigs *via* the Salmon tool in the output of the Eukrhythmic pipeline to the mapping to ORFs reported by nf-core/metatdenovo, and found general agreement between the rank order of number of reads mapped using both approaches (Table 2 and Table S4).

We also evaluated the consistency of the taxonomic annotations from the nf-core/metatdenovo pipeline compared to the multi-assembler approach mapped using Salmon within the Eukrhythmic pipeline (Fig. 8A). In general, the major taxonomic groups as assembled and quantified by nf-core/metatdenovo showed high relative abundance correlations with the same samples and taxonomic groups from Eukrhythmic. The proportion of sequences in the "Other" category (not annotated as a member of the major taxonomic groups) shows the highest discrepancy, while the general diatom class from nf-core/metatdenovo had higher relative abundance than it did in the Eukrhythmic results (Fig. 6).

Functional annotations from nf-core/metatdenovo were also consistent with the output of Eukrhythmic (Fig. 8B). Summing read counts assigned to transcripts assigned with KEGG orthology groups resulted in a regression line with slope 0.78 and an $R^2$ value of 0.7 ($p < 2.2e-16$; Fig. 8B).

## DISCUSSION

In the implementation of nf-core/metatdenovo, we decided to rely on Nextflow and the nf-core community. This was because domain-specific languages that act as workflow managers, such as Nextflow, reduce barriers to adoption of advanced, multi-step pipelines
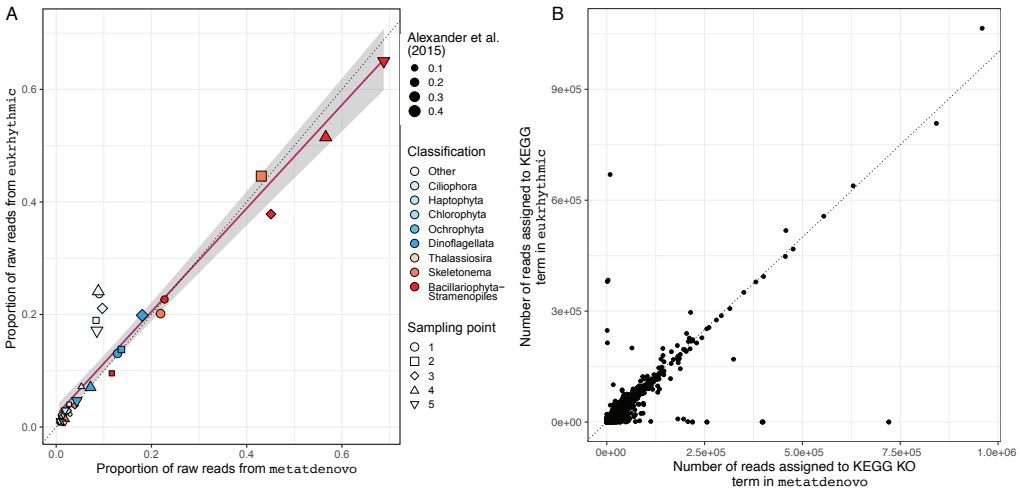

**Figure 8  Taxonomic and functional annotation comparison.** (A) Taxonomic annotation comparison. Proportion of raw reads assigned to each taxonomic group across the five temporal sampling points as determined by eukrhythmic (*y*-axis) compared to nf-core/metatdenovo (*x*-axis). Colors denote taxonomic groups while shapes denote sampling points; point size is scaled to the proportion of the raw reads assigned to each taxonomic group using the direct mapping approach used in *Alexander et al. (2015)* (B) Functional annotation comparison. Number of reads assigned to each KEGG orthology group in nf-core/metatdenovo (*x*-axis) *vs.* eukrhythmic (*y*-axis). Reads assigned to transcripts containing each KEGG annotation are summed; points indicate summed abundance of KEGG orthology groups in individual samples.

by providing access to multiple hardware platforms in a single interface (*Di Tommaso, Chatzou & Floden, 2017*). Domain-specific languages are often combined with container technologies that provide software without the need for additional installation, such as Docker and Singularity. Adoption of these workflows enables biologists simplified access to pipelines written by domain experts in bioinformatics. Nextflow pipelines can be executed from a UNIX-style command line environment without prior installation of pipeline code or tool-specific software. Configuration options tailor execution to different hardware platforms and software download and installation systems. Furthermore, the Seqera platform, a web-based graphical user interface, allows users to easily launch, manage, and monitor Nextflow pipelines and computing environments locally or on the cloud.

A considerable problem for users of published pipelines is that continued maintenance is not guaranteed, and thus presents a barrier to uptake. Larger collaborations have been formed to deal with this problem by providing a larger community of developers with an interest in collectively maintaining a set of pipelines (*Yamrom et al., 2022*). The nf-core collaboration maintains relatively strict code standards and retains a set of tools to ease the enforcement of standards, all to simplify maintenance and to distribute to various developers within the collaboration (*Ewels et al., 2019*). In line with this, it is common in nf-core that pipelines have several contributors. By providing the nf-core/metatdenovo pipeline as a part of the nf-core community, we hope to increase the chances for the
pipeline to evolve with the appearance of new software that can benefit annotation of metatranscriptomes.

The nf-core/metatdenovo pipeline was designed for the purpose of annotating metatranscriptomes. However, other types of data can be efficiently analyzed with the pipeline. For instance, metagenomes of bacteria, eukaryotes, and viruses can be assembled, ORF-called, and annotated using the tools in the pipeline. What nf-core/metatdenovo does not do is to generate Metagenome Assembled Genomes (MAGs). (To reconstruct MAGs, the nf-core/mag pipeline (*Krakau et al., 2022*) is an alternative.) Instead, when used for metagenomes, it will produce quantified and annotated gene catalogs. MAGs are important tools in meta-omics as they allow genome-resolved annotation of genes. For the foreseeable future, however, we believe that *de novo* assembly of both metatranscriptomes and metagenomes will be an important complement to genome-based annotations, particularly in very diverse microbial ecosystems.

Several other annotation pipelines exist for metatranscriptomes, one of which, Eukrhythmic, was used in the comparison of the Narragansett Bay dataset above. A detailed comparison of these tools with nf-core/metatdenovo to produce a recommendation is difficult to perform due to differences in overall approach, technology, and available annotation tools. Eukrhythmic (*Krinos et al., 2023*), a Snakemake pipeline, is the most sophisticated in terms of assembly as it allows the use of several assembly programs, and assembly of individual samples rather than all samples together, as nf-core/metatdenovo. The resulting assemblies are subsequently clustered into a single set of contigs. It is mostly targeted at eukaryotic metatranscriptomes by using TransDecoder, an ORF caller developed specifically for eukaryotic mature transcripts, also available in nf-core/metatdenovo. Eukrhythmic is set up to use EUKulele (*Krinos et al., 2021*)—developed in the same lab—for taxonomic annotation and the eggNOG-mapper for functional annotation, while nf-core/metatdenovo is set up to use additional annotation tools. The EUKulele tool is also available within nf-core/metatdenovo. The Snakemake-based design of Eukrhythmic requires the download of the pipeline code, and an initialization step that requires some familiarity with Snakemake. The nf-core/metatdenovo tool bypasses initialization *via* use of container technologies such as Docker or Singularity, making it more user-friendly, though the time and memory requirements of assembly tools continue to make integrated metatranscriptome assembly pipelines challenging to run contiguously. MetaGT (*Shafranskaya et al., 2022*) is another Nextflow pipeline based on *de novo* assembly, with an approach that uses both metagenomic and metatranscriptomic sequences to correct assembled sequences. As far as we understand, it quantifies transcripts but does not provide any annotation besides Prokka. It is currently at version 0.1.0, released in 2021. MetaPro (*Taj et al., 2023*) is a Python script bundled in a Docker or Singularity image. The script uses checkpointing to allow restart from intermediate results. As it's run as a single process, it does not have the scalability of Nextflow by being able to instantiate an arbitrary number of jobs on, for instance, HPC clusters or cloud infrastructure. Like the previously mentioned pipelines, it is based on *de novo* assembly—using rnaSPAdes—and performs quantification and a number of annotation steps. For annotation it uses standard tools like DIAMOND (*Buchfink, Xie & Huson, 2015*) and public databases such as NCBI's NR

(*Sayers et al., 2024*), and pipeline-specific summarization tools. The latest release was 3.0.1 in May 2023. SAMSA2 (*Westreich et al., 2018*) is also a Python script that uses script-specific checkpointing to allow restart in case of interruptions. However, it does not perform *de novo* assembly but is based on aligning raw reads to reference databases, a completely different approach to nf-core/metatdenovo and the other pipelines discussed here.

By developing our pipeline within the nf-core collaboration (*Ewels et al., 2019*), we're helped by the community and the tools developed in the collaboration to deliver a robust, well-documented and, hopefully, long-lived pipeline. Recently, a special interest group focusing on tools for microbial ecology—"meta-omics"—was formed. Within this group, we plan to develop training material, improve the interoperability of nf-core pipelines, and foster collaboration between nf-core developers and users of pipelines. One concrete example of improved interoperability is an ongoing effort to chain pipelines, so that output from nf-core/metatdenovo can easily be passed to *e.g.*, nf-core/differentialabundance to identify differentially abundant genes, nf-core/funcscan for deeper functional annotation, or nf-core/phyloplace for phylogenetic classification of genes.

## CONCLUSIONS

We have developed nf-core/metatdenovo to provide a robust and comparatively easy-to-use pipeline for *de novo* assembly-based annotation of sequence and metatranscriptomic data, as well as metagenomic, from complex communities. While the ease of use of Nextflow pipelines and the ability to run on various hardware platforms is attractive to end users. The nf-core community provides a helpful environment both for users of pipelines as well as potential future contributors fostering collaboration that hopefully leads to long-term maintenance of the pipeline.

## ACKNOWLEDGEMENTS

We thank the very helpful nf-core community for all the work provided in the creation of tools and the provision of discussion forums. In particular we would like to thank the following members for their contribution of code and review of pull requests: Daniel Straub, Mahesh Binzer-Panchal, James Fellow Yates, Jasmin Frangenberg, Friederike Hanssen and Taylor Falk. We thank Harriet Alexander for her contribution of data and feedback on early iterations of the project. Finally, we thank Nathan Van Wyk for his careful proofreading of the manuscript.

### Funding

The PhD studies of Danilo Di Leo were funded by the Linnaeus University. This work was supported by the Swedish Research Council VR Swedish Biodiversity Data Infrastructure (SBDI), project-IDs: 2019-00242 and 2023-00184. Arianna Krinos was supported by the U.S. Department of Energy, Office of Science, Office of Advanced Scientific Computing Research, Department of Energy Computational Science Graduate Fellowship under

Award Number DE-SC0020347. The funders had no role in study design, data collection and analysis, decision to publish, or preparation of the manuscript.

## Grant Disclosures

The following grant information was disclosed by the authors:

Linnaeus University.

Swedish Research Council VR Swedish Biodiversity Data Infrastructure (SBDI): 2019-00242, 2023-00184.

U.S. Department of Energy, Office of Science, Office of Advanced Scientific Computing Research, Department of Energy Computational Science Graduate Fellowship: DE-SC0020347.

## Competing Interests

The authors declare there are no competing interests.

## Author Contributions

- Danilo Di Leo conceived and designed the experiments, performed the experiments, analyzed the data, prepared figures and/or tables, authored or reviewed drafts of the article, and approved the final draft.
- Emelie Nilsson conceived and designed the experiments, performed the experiments, analyzed the data, authored or reviewed drafts of the article, and approved the final draft.
- Arianna Krinos performed the experiments, analyzed the data, prepared figures and/or tables, authored or reviewed drafts of the article, and approved the final draft.
- Jarone Pinhassi conceived and designed the experiments, authored or reviewed drafts of the article, and approved the final draft.
- Daniel Lundin conceived and designed the experiments, performed the experiments, analyzed the data, prepared figures and/or tables, authored or reviewed drafts of the article, and approved the final draft.

## DNA Deposition

The following information was supplied regarding the deposition of DNA sequences:

The raw sequence data are available at NCBI BioProject:

– Marine Bacterioplankton dataset: PRJEB10237

- Kimchi microbiome datasets: SRX128705, SRX128704, SRX128702, SRX128700, SRX128699

– Eukaryotic datasets: SRR1810799, SRR1810204, SRR1810801, SRR1810205, SRR1810206

## Data Availability

The Code is available at GitHub and Zenodo:

– Available at https://github.com/LNUc-EEMiS/metatdenovo-paper

– Daniel Lundin. (2025). LNUc-EEMiS/metatdenovo-paper: Paper publication (1.0). Zenodo. https://doi.org/10.5281/zenodo.17361583.

The Nf-core/metatdenovo pipeline is available at GitHub and Zenodo:

– https://github.com/nf-core/metatdenovo.

– Danilo Di Leo, & Emelie Nilsson & Daniel Lundin. (2025). github.com/nf-core/metatdenovo (1.1.0). Zenodo. https://doi.org/10.5281/zenodo.14927825.

The raw sequence data are available at NCBI BioProject:

– Marine Bacterioplankton dataset: PRJEB10237

- Kimchi microbiome datasets: SRX128705, SRX128704, SRX128702, SRX128700, SRX128699

– Eukaryotic datasets: SRR1810799, SRR1810204, SRR1810801, SRR1810205, SRR1810206

## Supplemental Information

Supplemental information for this article can be found online at http://dx.doi.org/10.7717/peerj.20328#supplemental-information.

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
