# Peer review of "The Nextflow nf-core/metatdenovo pipeline for reproducible annotation of metatranscriptomes, and more"

_PeerJ, doi:10.7717/peerj.20328_

## Round 0.1 · original submission · Major Revisions

· Academic Editor

Major Revisions

All reviewers agree that the manuscript presents a relevant and timely contribution, and most of the requested changes are relatively minor. However, a key concern was raised regarding the practical usability of the pipeline. Specifically, a reviewer reported that the pipeline does not function as smoothly "out of the box" as claimed by the authors. This discrepancy between the presented ease of use and the actual implementation experience needs to be thoroughly addressed. I would therefore request more extensive testing and clearer documentation. Resolving these issues is essential to ensure that the tool can be reliably used by the broader community.

Reviewer 1 ·

Basic reporting

One point I have to add in this regard is that, despite the article focusing on metatranscriptomes, the introduction of the paper is generic to metagenomics. The authors should expand the introduction - and maybe the discussion section - a bit, by highlighting what differentiates metatranscriptome analysis from standard metagenomics, and which steps cannot be handled by a standard metagenomics pipeline, and require the development of a discrete metatranscriptomics workflow.

Apart from the above, the article is mostly well-written in good English. Some syntax errors do exist; however, which can be fixed by having the manuscript proofed by a competent English speaker.

Experimental design

No comment

Validity of the findings

No comment

·

Basic reporting

This paper describes a pipeline that is part of the community effort led by the nf-core organisation, to provide a set of Nextflow-based pipelines that can easily adapt to different setups (clusters, public cloud, local servers...) and experimental designs. Metatdenovo fits in the scheme of providing a single pipeline per task, but allowing the user, via configuration, to customise the protocol or tools of choice.

Metatranscriptome sequencing is an increasingly popular technique used to obtain both taxonomical composition and coding potential of microbial communities. This paper adequately presents the workflow of the pipeline, and cites the relevant nf-core framework that provides a solid background on the technical foundations.

The pipeline is tested against published datasets, highlighting both the usefulness of this out-of-the-box solution and sometimes the challenges in replicating the exact experimental setup of original studies (e.g. Bunse et al. performed two per-sample assemblies), and even key results. I agree with the authors that this 'highlights the importance of reproducible annotation workflows'.

Some minor comments:

* The title misspells the pipeline (metadenovo should be metaTdenovo [capital for emphasis])
* In the abstract, I would not emphasise that the pipeline can be helpful for groups "lacking computational expertise", but instead that any group benefits from standardised and reproducible pipelines.
* Considering this will become the reference paper for the pipeline, I would expand the introduction with what a metat denovo workflow entails, with sub-paragraphs linked to the "pipeline stages": Pre-processing, Assembly, Mapping, Gene annotation, Quantification, Taxonomy & functional annotation, Pipeline summary

Experimental design

The output of the authors is a pipeline that any user can easily deploy and use on their own dataset. The pipeline is composed by well known tools and there is no intent of justifying the selection of tools or claim to outperform existing best-practices.

The re-analysis of existing dataset is mostly a showcase of the ease of use of the pipeline rather than an assessment, lacking a ground truth.

The pipeline itself is an excellent example of publicly available, user-contributed open source pipeline, and sufficient documentation is provided from the nf-co.re website.

In the spirit of nf-core, I would suggest rewriting the "Statistical analysis" paragraph, as it's of no interest in itself to discuss the specific version of R with no other information. I would recommend briefly describing the analyses performed and providing links to versioned R scripts used for the study, possibly putting in the repository also the raw data analysed to allow re-creating the plots.

Validity of the findings

no comment

Additional comments

Figure 2 would be more readable if all the labels were enlarged.

I would encourage checking the palette used for figures (esp. Figure 7c) to be color-blind friendly.

·

Basic reporting

The manuscript is well-structured and fairly easy to follow. The language is good and only minor corrections are needed. Literature references are relevant and sufficient.
The text is supported by a substantial amount of graphical material. The quality of some plots can be improved.

See full report in PDF attached.

Experimental design

Since this is a tool paper, there is not much of experimental design. Methods are described with sufficient details, choice of the software is reasonable.
The authors demonstrate different aspects of the developed pipeline using 3 completely different datasets and compare their results to the original studies, which in my opinion is sufficient.

See full report in PDF attached.

Validity of the findings

Similarly, "validity of findings" is not entirely relevant for software papers. All used data is available and the the pipeline can be replicated by its design.

See full report in PDF attached.

Additional comments

See full report in PDF attached.

---

## Round 0.2 · accepted · Accept

· Academic Editor

Accept

Both reviewers were satisfied with your revisions and I am happy to inform you that your manuscript is now accepted for publication. I thank you and also the reviewers for their time and effort you put into this process!

Reviewer 1 ·

Basic reporting

no comment

Experimental design

no comment

Validity of the findings

no comment

Additional comments

The authors have addressed my points, as well as the points of the other reviewers. The quality and content of the text has been considerably improved. I therefore recommend that this manuscript be accepted for publication.

·

Basic reporting

I thank the authors for addressing all my comments. I hope that this revision improved both the manuscript and the software itself.

The manuscript is well-structured and easy to follow, the authors fixed all minor issues related to plots, structure and wording. Literature references are sufficient.

Experimental design

(Remains unchanged from previous revision - all good)
Since this is a tool paper, there is not much of experimental design. Methods are described with sufficient details, choice of the software is reasonable.
The authors demonstrate different aspects of the developed pipeline using 3 completely different datasets and compare their results to the original studies, which in my opinion is sufficient.

Validity of the findings

(Remains unchanged from previous revision - all good)
Similarly, "validity of findings" is not entirely relevant for software papers. All used data is available and the the pipeline can be replicated by its design.